# A Pilot Study towards the Impact of Type 2 Diabetes on the Expression and Activities of Drug Metabolizing Enzymes and Transporters in Human Duodenum

**DOI:** 10.3390/ijms20133257

**Published:** 2019-07-02

**Authors:** Sophie Gravel, Benoit Panzini, Francois Belanger, Jacques Turgeon, Veronique Michaud

**Affiliations:** 1Faculty of Pharmacy, Université de Montréal, Montreal, QC H3C 3J7, Canada; 2CRCHUM—Centre de Recherche du Centre Hospitalier de l’Université de Montréal, Montreal, QC H2X 0A9, Canada; 3CHUM—Centre Hospitalier de l’Université de Montréal, Department of Gastroenterology, Montreal, QC H2X 0A9, Canada; 4Faculty of Medicine, Université de Montréal, Montreal, QC H3C 3J7, Canada

**Keywords:** drug metabolism, cytochromes P450, intestine, type II diabetes mellitus

## Abstract

To characterize effects of type 2 diabetes (T2D) on mRNA expression levels for 10 Cytochromes P450 (CYP450s), two carboxylesterases, and three drug transporters (ABCB1, ABCG2, SLCO2B1) in human duodenal biopsies. To compare drug metabolizing enzyme activities of four CYP450 isoenzymes in duodenal biopsies from patients with or without T2D. mRNA levels were quantified (RT-qPCR) in human duodenal biopsies obtained from patients with (*n* = 20) or without (*n* = 16) T2D undergoing a scheduled gastro-intestinal endoscopy. CYP450 activities were determined following incubation of biopsy homogenates with probe substrates for CYP2B6 (bupropion), CYP2C9 (tolbutamide), CYP2J2 (ebastine), and CYP3A4/5 (midazolam). Covariables related to inflammation, T2D, demographic, and genetics were investigated. T2D had no major effects on mRNA levels of all enzymes and transporters assessed. Formation rates of metabolites (pmoles mg protein^−1^ min^−1^) determined by LC-MS/MS for CYP2C9 (0.48 ± 0.26 vs. 0.41 ± 0.12), CYP2J2 (2.16 ± 1.70 vs. 1.69 ± 0.93), and CYP3A (5.25 ± 3.72 vs. 5.02 ± 4.76) were not different between biopsies obtained from individuals with or without T2D (*p* > 0.05). No CYP2B6 specific activity was measured. TNF-α levels were higher in T2D patients but did not correlate with any changes in mRNA expression levels for drug metabolizing enzymes or transporters in the duodenum. T2D did not modulate expression or activity of tested drug metabolizing enzymes and transporters in the human duodenum. Previously reported changes in drug oral clearances in patients with T2D could be due to a tissue-specific disease modulation occurring in the liver and/or in other parts of the intestines.

## 1. Introduction

Type 2 diabetes (T2D) is the most common form of diabetes mellitus worldwide and its global prevalence is increasing yearly [1]. Patients with T2D often require multiple drugs to treat their numerous comorbidities. They also present a highly variable response to drugs: some T2D patients appear resistant to drugs while others are more sensitive (e.g., warfarin, clopidogrel, cyclosporine, tacrolimus, and anti-hypertensive agents) [2,3,4,5,6,7]. The exact mechanism underlying this interindividual variability in drug response is still unknown.

Inter-subject variability in drug response is largely explained by drug pharmacokinetics and pharmacodynamics (PK-PD). First pass elimination of orally administered drugs involves both gastro-intestinal and hepatic-mediated metabolism and transport. In addition to the liver, the small intestine could be a major contributor to pre-systemic drug-metabolism and excretion (through efflux transporters).

Cytochrome P450s (CYP450s) constitute a superfamily of enzymes contributing largely to the metabolism of drugs. CYP450s are highly expressed in tissues involved in first pass metabolism such as the intestines and liver. CYP450 expression and activity profiles are tissue specific and, clearly, extra-hepatic tissues differ from the liver [8,9,10,11,12,13]. Various expression profiles for drug metabolizing enzymes and transporters have been observed along the intestinal tract [14,15,16,17]. There is a general consensus that the most highly expressed CYP450s in human intestines are CYP2C and CYP3A subfamilies [18].

CYP450 metabolic activity can be regulated by numerous factors including genetic makeup, environmental factors, concomitant medications, and inflammation [19,20,21,22,23,24,25]. For instance, inflammatory mediators, such as cytokines, can modulate CYP450-dependent drug metabolism [20,26,27,28,29]. T2D pathophysiology is associated with a chronic active inflammatory status and patients with T2D also present an important prevalence of metabolic syndrome that is, in turn, linked to a status of low-chronic inflammation [30,31,32].

Using a diet-induced obesity model of T2D in mice, our laboratory has demonstrated that T2D can affect CYP450 expression and activities in an isoenzyme and tissue-specific manner [12]. In a clinical case-control pharmacokinetic study, our preliminary results showed that patients with T2D presented a decreased metabolic activity for CYP2B6, CYP2C19, and CYP3As following the oral administration of a cocktail of probe drugs [33,34]. However, very few reports (if any) are available on the impact of T2D on human intestinal CYP450s. In animal models (mice and rats), expression of cyp3a in the small intestine was shown to be altered by diet-induced or streptozotocin-induced diabetes [35,36,37].

Our aim was to investigate whether intestinal CYP450 activities are modulated by T2D in an isoform-specific manner by assessing major CYP450 expression and activities in human duodenal biopsies from patients with or without T2D. As a secondary objective, the impact of T2D on other drug metabolizing enzymes and transporters was examined. Additionally, the influence of inflammation, genetic, demographic, and T2D-related variables on major CYP450 activities was determined.

## 2. Results

As presented in Table 1, study population was composed of 16 non-diabetic participants and 20 patients with T2D. Individuals with T2D had a slightly higher BMI compared to patients without T2D (*p* = 0.03). Biomarkers of T2D (including insulin levels, glycemia, HbA1C, and HOMA-IR) and drugs used in this condition were all significantly higher in the T2D group, as expected per protocol inclusion criteria. Most of the participants enrolled in our study were Caucasians (*n* = 29) while two subjects were Blacks and one Asian (data was missing for four individuals).

As illustrated in Figure 1A,B, the most expressed mRNAs in both study groups for drug metabolizing enzymes were CES-2, CYP2C9, and CYP3A. Figure 1C,D show that the most abundant drug transporters in the duodenum for non-T2D and T2D groups was ABCG2, which is followed by OATP2B1 and ABCB1.

Expression profile for all drug metabolizing enzymes and transporters was similar between the two study groups (Figure 1). As presented in Table 2, T2D had no effect on mRNA levels for all studied drug metabolizing enzymes (CYP450s and CES) (*p*-value > 0.05), even though a tendency was observed with increased CYP2C9 mRNAs (*p* = 0.051) in patients with T2D. When adjusting for age and gender in the multivariate regression model analyses, the influence of diabetes on expression levels of studied drug metabolizing enzymes remained insignificant (adjusted *p*-value > 0.05, Table 2). The expression profile for one drug transporter, known as OATP2B1, seemed to differ between individuals with T2D compared to non-diabetic patients independently of age and gender. As shown in Table 2, levels of mRNA transcripts for OATP2B1 were slightly higher, about 20%, in patients with T2D than in the non-diabetic controls (*p*-value and adjusted *p*-value = 0.02). However, this could be of marginal clinical relevance. Illustrative graphs of mRNA transcript levels for all drug metabolizing enzymes and transporters in all patients with T2D vs. non-diabetic controls are displayed in Appendix A.

In this study, sampling of duodenal biopsies enabled the determination of activity levels for four important CYP450 isozymes, which are CYP2B6, CYP2C9, CYP2J2, and CYP3A4/5. Both study groups exhibited activities for CYP2C9 (hydroxytolbutamide), CYP2J2 (hydroxyebastine), and CYP3A4/5 (1′-hydroxymidazolam), but not for CYP2B6 (hydroxybupropion). No significant difference was measured for the formation rate of the various metabolites (mean ± SD) between individuals without T2D or patients with T2D for CYP2C9 (0.41 ± 0.12 vs. 0.48 ± 0.26 pmoles mg protein^−1^ min^−1^), CYP2J2 (1.69 ± 0.93 vs. 2.16 ± 1.70 pmoles mg protein^−1^ min^−1^) or CYP3A (5.02 ± 4.76 vs. 5.25 ± 3.72 pmoles mg protein^−1^ min^−1^) (*p*-value > 0.05, Figure 2). When controlling for age, the gender and genotype metabolizer status with diabetic status as the dichotomous predictor in a linear regression model, showed no significant effect on activity levels of all studied isoforms, which was revealed. Moreover, no significant correlation was observed between relative mRNA levels and metabolic activity for CYP2C9, CYP2J2, and CYP3A4/5 (Appendix A).

To confirm that the measured metabolic transformation of probe substrates used was due to the targeted CYP450 pathway. In vitro incubations in the presence of specific inhibitors were performed. Activity levels for CYP2J2 and CYP3A in the presence of astemizole and ketoconazole were reduced by 89 ± 5% and 93 ± 5%, respectively. Formation rate of hydroxytolbutamide in the presence of sulfaphenazole was decreased to levels below or near the level of quantification (5 nM hydroxytolbutamide). For CYP2B6, the trivial (although detectable) hydroxybupropion formation was not inhibited by the selective inhibitor ticlopidine. This suggests that the metabolic activity observed is not specific and that CYP2B6 is not a major contributor to bupropion metabolism in the duodenum.

As reported in Table 3, mean ± SD levels of TNF-α cytokine were significantly (*p* = 0.03) higher in blood samples from individuals with T2D (2.71 ± 1.25 pg mL^−1^) compared to the non-diabetic patients (2.00 ± 0.36 pg mL^−1^). However, no significant difference between non-diabetic and T2D patients was measured for the other inflammatory markers (IFN-γ, IL-1β, and IL-6). We pursued our analysis by trying to measure a correlation between those cytokine levels with measured metabolic activities of CYP2C9, CYP2J2, and CYP3A. Results reported in Appendix A show no correlation between quantified proinflammatory cytokines and measured activities for the various CYP450 isoforms (Appendix A).

Influence of numerous covariables on CYP2C9, CYP2J2, and CYP3A activities such as T2D-related covariables (insulinemia, glycemia, HbA1c, and HOMA-IR) and demographic covariables (age and BMI) was tested. Only one significant correlation was found between HbA1c and the hydroxytolbutamide formation rate (*r*_s_ = 0.35 and *p* = 0.04, Appendix A). Influence of gender on CYP2C9, CYP2J2, and CYP3A activities was considered and no significant effect was observed (*p* = 0.8, 0.3, and 0.4, respectively). Time since diagnosis of T2D could also be a covariable affecting development of comorbidities and inflammation, which could, in turn, affect CYP450 activities. Hence, we divided our diabetic population into three groups (less than 5 years, 5 to 10 years, and more than 10 years) and no statistically significant effect on time since diagnosis on metabolic activities was observed (Appendix A).

Lastly, a qualitative evaluation (because of the small number of subjects in some genotyping groups) of the influence of CYP450 genotypes on metabolic activity was performed for the entire study population. Individuals homozygous for variants *CYP2C9*2* or *CYP2C9*3* presented lower mean values of metabolic activities than their wild type or heterozygotes counterparts. For *CYP2J2*, the presence of a **7* variant reduced mean metabolic activity by almost 50% compared to wild type *CYP2J2*1/*1* (1.04 vs. 2.07 pmol·mg·protein^−1^·min^−1^). Mean metabolite formation rates for individuals carrying one **22* allele for *CYP3A4* were decreased when compared to wild type individuals. Lastly, the formation rate of 1′-hydroxymidazolam was slightly lower in non-expressers vs. expressers of the *CYP3A5* gene (Appendix A).

## 3. Discussion

In this report, we show that T2D has no major impact on mRNA expression levels of 10 major intestinal CYP450 isoforms, two carboxylesterases, and three transporters as measured in duodenal biopsies obtained from patients with or without T2D. Furthermore, the metabolic activity of three important CYP450s namely, CYP2C9, CYP2J2, and CYP3As, were not modulated significantly by T2D.

We measured significant mRNA amounts for multiple CYP450 isoforms with mRNAs of CYP2C9 and CYP3A4/5 being the most expressed. This is in agreement with several studies reporting mRNA levels or immune-quantified proteins of CYP450s in the human small intestine [16,17,18,38,39,40]. High levels of mRNA for CES-2 in duodenal biopsies for both study groups were also detected in this study. It is known that CES are highly expressed in liver and intestines and, particularly, CES-2 as the major isozyme in the small intestine [41,42,43].

Duodenal mRNA expression of all CYP450s and CES was similar between individuals with or without T2D. Very few studies have looked at the impact of T2D on CYP450 expression in humans and, if so, most studies focused on the liver. One study reported an increased protein expression of CYP2E1 in human liver microsomes from diabetic patients even though mRNA levels were similar to non-diabetic controls [44]. Another study in humans investigated the impact of T2D on CYP2E1 [44]. This study reported that CYP2E1 mRNA measured in peripheral blood mononuclear cells was increased in patients with T2D compared to the non-diabetic control [44]. In our study, relative mRNA levels CYP2E1 were very low in both study groups (<0.04%) and no difference was observed between the two groups. This result is in agreement with several studies reporting low or undetectable levels of CYP2E1 in the small intestines [18,45,46,47,48]. CYP3A is another CYP450 family for which mRNA transcripts have been quantified in individuals with T2D. Again, although a decreased protein expression and metabolic activity for CYP3A in human liver microsomes from diabetic patients was observed, this change was not reflected in mRNA levels [49]. We also did not observe any effects of T2D on mRNA levels for CYP3A in duodenal biopsies.

Some studies report on the impact of T2D on CYP450 mRNA levels using animal models [12,35,50,51,52,53]. Available reports in mice models display highly variable results and most observations are on hepatic cyp450s even though significant changes in mRNA levels for cyp1a, cyp2a, cyp2b, cyp2c, cyp2e, and cyp3a were reported in the liver, kidney, and lungs [12,35,50,51,52,53]. Results are often contradictory and no clear conclusion on the effects of diabetes on cyp450s mRNA levels can be drawn.

The presence of three drug-transporter mRNAs (ABCB1, ABCG2, and OATP2B1) in duodenum biopsies was detected and levels of OATP2B1 were slightly higher in the T2D group. Previous animal studies in mice showed no impact of diabetes on abcg2, Mrp2, and Mrp3 in the liver [51,53]. Results pertaining to the influence of diabetes on ABCB1 are variable. Some data are available in the intestines, but one group reports a decrease in duodenal expression for a diabetic model in mice while another study shows an increase in intestinal expression using an obese rat model [54,55].

CYP3A is involved in the metabolism of about 50% of all drugs and is highly expressed in both the liver and intestines [18,56,57]. A CYP3A metabolic activity was measured in the S9 fractions (5.15 pmol of 1′-hydroxymidazolam mg·protein^−1^·min^−1^) prepared from duodenum biopsies of our subjects. Important CYP3A-dependent midazolam metabolism in human intestines has been reported previously in microsomal fractions even though the contribution of intestinal CYP3As to midazolam metabolism is less than CYP3A intrinsic clearance in the liver by about 10-fold [58,59,60]. One study reported mean CYP3A activity of 230 pmol of 1′-hydroxymidazolam mg·protein^−1^·min^−1^ in pooled microsomes obtained from the entire human intestine [60]. Other studies reported significant CYP3A midazolam metabolism in duodenal homogenates from healthy subjects with a mean hydroxymidazolam formation rate of 67 to 446 pmol·mg·protein^−1^·min^−1^ [61,62,63]. We recently found similar CYP3A metabolic activities in intestinal microsomes from human duodenum with 1′-hydroxymidazolam formation ranging from one to 213 pmol·mg·protein^−1^·min^−1^ (mean 69 ± 75) [17].

In a diabetic mice model, intestinal CYP3A metabolic activity was shown to increase compared to controls [35]. In humans, we previously demonstrated a decreased oral clearance of the CYP3A probe substrate midazolam in patients with T2D compared to non-diabetic subjects [34]. In this study, we observed no effect of T2D on duodenal CYP3A activity in vitro (*p* = 0.47). This suggests that the reduced oral clearance of midazolam that we observed previously in T2D patients could be explained by a tissue-specific modulation of CYP3A occurring most likely in the liver. This hypothesis is supported by other studies demonstrating a decrease in 1′-hydroxymidazolam formation in human liver microsomes from patients with T2D [49]. Another explanation why no difference was found in this study can rely on the patient’s pre-medical conditions: i.e., the gastro-intestinal endoscopy was medically indicated. Thus, the pre-medical conditions could have already induced a phenol conversion in their duodenum and the impact of other concomitant pathologies such as diabetes is blunted and could not be revealed.

CYP2C9 is an important isoform of the CYP450s both in the liver and in the small intestines [18,58]. It is noteworthy that multiple oral antidiabetic drugs such as sulfonylureas, meglitinides, and thiazolidinediones are metabolized by CYP2Cs [64,65]. Significant metabolic activity for CYP2C9 in human intestines has been reported even though it was found to be lower than CYP3A activities. Our results are in agreement with these observations [5.15 pmol of 1′-hydroxymidazolam mg·protein^−1^·min^−1^ (CYP3A) vs. 0.45 pmol of hydroxytolbutamide mg·protein^−1^·min^−1^ (CYP2C9)] [39,60]. Using tolbutamide as a probe substrate, two studies reported mean activity levels of 3 and 13 pmoles of hydroxytolbutamide mg·protein^−1^·min^−1^ [17,60]. These values are slightly higher than those measured in our biopsy S9 fractions, which is a diluted matrix compared to microsomes. Additionally, biopsies were obtained from non-healthy non-diabetic and diabetic patients for whom a gastro-intestinal endoscopy was medically indicated. Our results suggest that T2D had no impact on CYP2C9 activity locally in the human duodenal biopsies (*p* = 0.4). This finding is in agreement with a study that revealed no influence of diabetes on CYP2C activity in rats [66].

CYP2J2 expression and activity were investigated since this isoform has been reported to participate in the metabolic transformation of many drugs in addition to its important role in arachidonic acid transformation [67,68,69,70]. In this study, we measured a relatively low mRNA expression level for CYP2J2 in the duodenum (0.4% of all CYP450s quantified). However, its expression was constant in all participants with a low interindividual variation. It has been reported before that CYP2J2 expression is constant throughout the entire gastrointestinal tract, but its low immune-quantified CYP protein (0.9 pmol·mg^−1^) raises doubt about its overall importance in the first-pass metabolism of drugs [18,71]. Despite its low expression, we measured significant levels of CYP2J2 specific activity in the duodenum using the probe substrate ebastine, which is a selective probe for CYP2J2 over CYP3As [69,72]. This is in agreement with two previous studies reporting metabolic activities in human intestinal microsomes or in human duodenal microsomes with activity levels, respectively, which ranges from 0.009–0.076 nmol·mg^−1^·min^−1^ and averages 0.026 nmol·mg^−1^·min^−1^, respectively, compared to the 0.0006–0.007 nmol·mg^−1^·min^−1^ obtained for this study [17,69].

In the last few years, CYP2J2 has been studied for its possible implication in T2D pathophysiology and its role in comorbidities linked to its implication in arachidonic acid metabolism to epoxyeicosatrienoic acids (EETs). The CYP2J2-EETs-sEH metabolic pathway has been shown in mice to have a beneficial effect on adiposity, non-alcoholic fatty liver disease, systemic inflammation, and insulin resistance [73,74,75]. In two distinct studies, endothelial and cardiovascular tissue-specific overexpression of CYP2J2 in diabetic mice reduced nephropathy and cardiomyopathy, which are important diabetic comorbidities [76,77]. In our study, levels of CYP2J2 activity in subjects affected with diabetes were similar to patients without metabolic syndromes.

CYP2B6 mRNA relative level of expression in the duodenum were low (0.6%) in both study groups. We tested CYP2B6 activity in duodenal biopsies from T2D and non-diabetic patients by incubating bupropion with duodenal biopsy homogenates present and absent of the CYP2B6 selective inhibitor ticlopidine. In the small intestine, no significant CYP2B6 specific activity was measured. This is in agreement with a previous study that showed that no hydroxybupropion, which is the CYP2B6 specific metabolite, was produced in intestines and that metabolism of bupropion in this extra-hepatic tissue seems to be through multiple carbonyl reductase enzymes [78].

Lastly, CES-2 mRNA levels were detected in duodenal biopsies. High expression levels of CES-2 have been reported previously [41,42,43]. Among transporters studied, the most abundant drug transporters in the duodenum was ABCG2, which is followed by OATP2B1 and ABCB1. This finding is in agreement with other studies reporting that ABCG2 exhibited high expression levels in the human duodenum [79,80].

Although non-parametric tests did not show significant effects of covariables on expression or activity, the influence of gender and age were controlled in the linear regression analysis models since those factors have been shown to influence liver activity and expression of CYP450s in an isoform-specific manner [81,82,83,84]. Concerning modulation of intestinal CYP450s expression and activity by age and gender, not much is available. For CYP3A, evidence suggests that gender might only influence CYP3A in the liver, but not in the intestines [85]. In the linear regression models with diabetes as a dichotomous variable, gender seemed to influence significantly CYP2C19 duodenal mRNA levels with higher expression in female subjects (β = 1.28, *p* = 0.01). No influence of gender on CYP3A and all other tested enzymes and transporters expression or activity was unveiled by the linear regression analysis models. This may suggest that modulation as a function of gender can be tissue, isoform, and/or substrate-dependent.

Our data suggest that CYP450 activities were not modulated by T2D in the duodenum. This finding is based on a small number of subjects and an important inter-subject variability was observed. Consequently, further investigations are needed to confirm our current findings and whether changes in reduced oral clearance observed clinically following oral drug administration in patients with T2D can be explained by a tissue-specific modulation occurring either in the liver or in other parts of the intestine. This report, alongside with previously reported results on the impact of T2D on CYP450 expression and activities, supports the hypothesis that T2D influence drug metabolism via CYP450 modulation in a tissue and isoform-specific manner in humans. This type of effect of T2D has been vastly explored in animal models but results remain highly variable and dependent on the model used. Results of comprehensive studies showing the modulation of multiple CYP450s expression and activity in various hepatic and extra-hepatic tissues of animal models by T2D have been reported recently [12,35,50,51,52,66,86]. However, only sparse data are available in humans and knowledge on the impact of T2D on CYP450s has to be gained to understand the high interindividual response to treatment observed in this sub-population.

## 4. Methods

### 4.1. Subjects

Study protocol (Trial #12.386) was approved by the ethic review board of the CHUM research center (CRCHUM, Montreal, Canada) (05/09/2013). This trial was carried out in compliance with the Declaration of Helsinki and the International Conference on Harmonization Good Clinical Practice Guidelines. Written informed consent was obtained from all participants prior to any initiation of the study procedure.

A total of 36 participants referred to the CHUM gastroenterology department for a scheduled gastrointestinal endoscopy were recruited to constitute two study groups. This includes a group of 16 subjects classified as non-diabetic volunteers, according to their medical histories, physical examinations, and relevant laboratory tests and a second group of 20 patients were diagnosed with T2D. All patients were required to be ≥18 years old and to abstain from grapefruit juice within two weeks before the digestive endoscopy. Exclusion criteria included use of CYP450 inhibitors or inducers, patients with altered renal or hepatic functions, presence of an important inflammatory condition, or a posteriori diagnosis of duodenitis.

### 4.2. Study Design and Sample Collection

On the morning of the gastrointestinal endoscopy, participants were admitted at the CHUM after an overnight fast and restrained from all medication (except for insulin) until procedures were over or as instructed by the gastroenterologist. Prior to procedures, five blood samples were drawn in Vacutainers^®^ tubes (Becton-Dickinson). Two 5 mL blood samples collected in SST II vials were sent to the CHUM biochemistry laboratory for insulin and glucose measurements and three separate blood samples collected in Vacutainers^®^ with EDTA were processed as follows: one tube was sent to the CHUM biochemistry laboratory to measure glycated hemoglobin and the other two blood samples were sent to the research laboratory to determine genotypes of studied CYP450s and to quantify proinflammatory cytokine levels (IFN-γ, IL-1β, IL-6, and TNF-α). During the intervention, eight to 10 biopsies (about 2 to 4 mg/specimen) were sampled in the small intestine from the second part of the duodenum downstream to the hepatopancreatic ampulla. Two biopsies were preserved for the conventional histologic analyses. Two other biopsies were snap-frozen in liquid nitrogen and preserved at −80°C for mRNA analysis. Six biopsies were conserved on ice in sterile normal saline and transferred immediately to the CRCHUM laboratory to serve for CYP450 activity analyses. Patients were discharged after the sample collection and gastrointestinal endoscopy.

### 4.3. Determination of Intestinal CYP450 activities

Upon reception of the biopsies in sterile normal saline at the laboratory facilities, biopsies were transferred into tubes containing 3 mL of a solution (pH 7.4) composed of TRIS 50 mM, KCl 150 mM, EDTA 1 mM, benzamidine 1 mM, and 50 µg·mL^−1^ aprotinin (glycerol 10%) similar to those previously described [62,63,87]. Tissue was then homogenized on ice. The resulting preparation was centrifuged at 4 °C and 10,000× *g* for 20 min and supernatant (S9 fraction) was snap-frozen in liquid nitrogen and kept at −80 °C until determination of CYP450 activities.

On study day, the resulting supernatants were thawed on ice and incubated with CYP450 probes: 600 µM bupropion (CYP2B6), 400 µM tolbutamide (CYP2C9), 5 µM ebastine (CYP2J2), and 10 µM midazolam (CYP3A4/5). Specificity of the probe substrates in the intestinal biopsies was evaluated using selective inhibitors: 10 µM ticlopidine (CYP2B6), 10 µM sulfaphenazole (CYP2C9), 150 µM astemizole (CYP2J2), and 5 µM ketoconazole (CYP3A4/5). Following incubation, the formation rate of metabolites was determined by measuring their concentrations using a sensitive and precise LC-MSMS detection method [12,88]. Activity was normalized for total protein content measured using the Pierce^®^ BCA Protein Assay Kit (ThermoFisher Scientific Inc., USA). Details on chemicals, incubation procedures, and detection methods have been reported elsewhere [12].

### 4.4. Quantification of Intestinal CYP450 mRNAs and CYP450 Genotype Analysis

Standard methods were used and described in Appendix A (method details).

### 4.5. Determination of Proinflammatory Cytokine Levels

Blood samples for the quantification of proinflammatory cytokines were kept on ice and rapidly sent to the research laboratory to be centrifuged at 1100× *g* (10 min at 4 °C) within 1 hour of sampling. Plasma was then aliquoted and stored at −80 °C until use. Levels of inflammatory markers INF-γ, IL-1β, IL-6, and TNF-α were quantified by electrochemiluminescence immunoassays using the V-PLEX Proinflammatory Panel 1 Human Kit, QuickPlex SQ120 Imager, and WORKBENCH software (MSD^®^, Rockville, MD).

### 4.6. Statistical Analysis

To control for sex, age, and genotype metabolizer status, we ran linear regression models with diabetes status as a dichotomous predictor of phenotypic activity and mRNA expression levels of studied CYP450s and other drug metabolizing enzymes or transporters using R Statistical Software version 3.5.1 (Foundation for Statistical Computing, Vienna, Austria). GraphPad Prism 5 (GraphPad Software, La Jolla, CA) was used to perform spearman rank correlation coefficient analyses with two-sided *p*-value and α < 0.05 to test the impact of continuous covariables (insulinemia, glycemia, HbA1c, HOMA-IR, HOMA-β, age, BMI, time since diagnostic and proinflammatory cytokine levels) on CYP450 activities and mRNA expression levels. Lastly, the impact of discrete covariables (genotype, gender, and time since diagnostic categories) on CYP450 activities and expression were evaluated using Mann-Whitney or Kruskall-Willis analyses on GraphPad Prism 5 (GraphPad Software, La Jolla, CA). When relevant adjustment for a false discovery rate using Benjamini-Hochberg multiple correction procedures were applied.

## Figures and Tables

**Figure 1 ijms-20-03257-f001:**
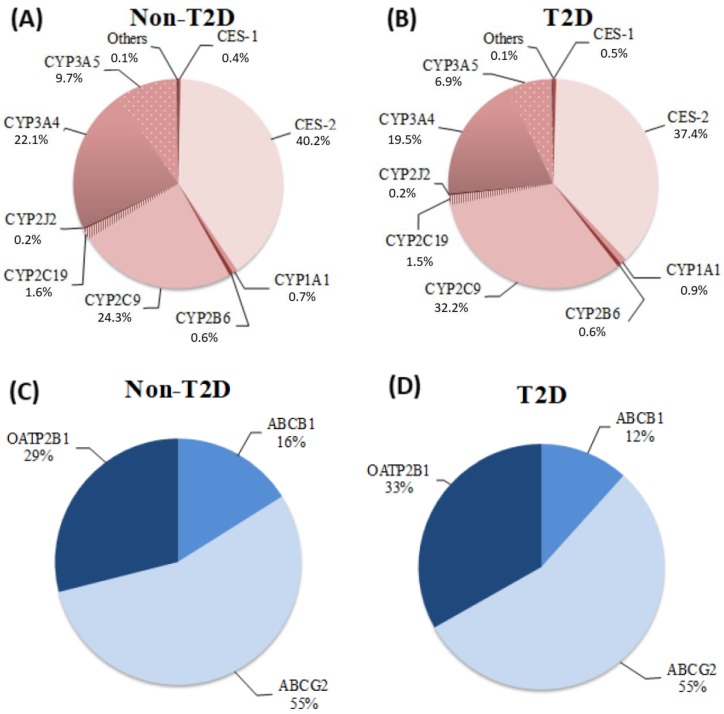
Drug metabolizing enzymes and transporters’ relative mRNA expression levels. Total mRNA transcripts (2^−^^Δ*C*T^) for each drug metabolizing enzymes (CYP450s and CES) and drug-transporters are displayed as expressed in human duodenal biopsies according to study group: (**A**) relative mRNA expression of drug metabolizing enzymes in non-T2D patients (*n* = 15). (**B**) Relative mRNA expression of drug metabolizing enzymes in T2D patients (*n* = 20). (**C**) Relative mRNA expression of drug-transporters in non-T2D patients (*n* = 15). (**D**) Relative mRNA expression of drug transporters in T2D patients (*n* = 20). CYP450 mRNA transcript with a relative contribution >0.2% are illustrated, and “others” have a relative contribution ≤ 0.07%. Others include the following isoforms: CYP2C8, CYP2D6, and CYP2E1.

**Figure 2 ijms-20-03257-f002:**
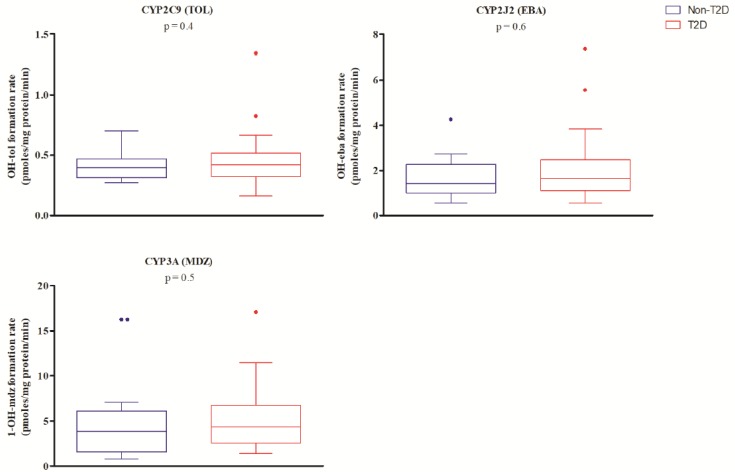
Biopsy homogenate (S9 fraction) activities. Rate of pathway-specific metabolite formation (pmoles mg protein^−1^ min^−1^) for CYP2C9 (Tolbutamide (TOL) → Hydroxytolbutamide (OH-tol)), CYP2J2 (Ebastine (EBA) → Hydroxyebastine (OH-eba)) and CYP3A (Midazolam (MDZ) → 1’-Hydroxymidazolam (1-OH-mdz)) in non-diabetic patients (non-T2D) and patients with T2D are presented as box plots with Tukey whiskers.

**Table 1 ijms-20-03257-t001:** Baseline demographic data and clinical characteristics of patients.

Parameters	Non-T2D Patients	Patients with T2D
No. of subjects	16	20
Sex: No. (%) M:F	7:9 (44:56)	8:12 (40:60)
Age (years)	57 ± 16	62 ± 11
BMI (kg m ^−2^)	25.8 ± 5.7	30.2 ± 6.6 *
Insulin (pmol L^−1^)	44.8 ± 26.2	188.2 ± 262.7 *
Glycemia (mmol L^−1^)	4.8 ± 0.5	7.1 ± 2.3 *
HbA1C (%)	5.4 ± 0.4	7.2 ± 1.0 *
HOMA-IR	1.4 ± 0.9	11.4 ± 25.8 *
HOMA-β	98 ± 50	151 ± 145
Time since diagnostic (years)	NA	9.5 ± 5.8
Medication use, No. (%) of subjects		
Metformin	0	13 (65) *
Sulfonylurea	0	7 (35) *
DPP4-I	0	3 (15)
Insulin	0	5 (25)
Statins	2 (13)	9 (45)
ACEI	2 (13)	4 (20)
ARB	2 (13)	7 (35)
CCB	2 (13)	4 (20)
PPI	10 (63)	8 (40)
β-Blockers	2 (13)	4 (20)
Aspirin	4 (26)	8 (40)
other NSAID	3 (19)	1 (5)
Antidepressants	1 (6)	5 (25)

Continuous variables are presented as mean ± SD. * Demographic parameters are significantly different between study groups (*p* < 0.05). NA, not applicable. Non-T2D, non-diabetic patient group. T2D, patients with a diagnostic of Type 2 diabetes group. BMI, body mass index. HbA1C, glycated hemoglobin. HOMA-IR, homeostatic model assessment of insulin resistance. HOMA-β, homeostatic model assessment of bêta cells function. DPP4-I, dipeptidyl peptidase-4 inhibitors. ACEI, angiotensin-converting-enzyme inhibitors. ARB, angiotensin II receptor blockers. CCB, calcium channel blockers. PPI, proton pump inhibitors. NSAID, non-steroidal anti-inflammatory drugs.

**Table 2 ijms-20-03257-t002:** mRNA transcripts for drug metabolizing enzymes and transporters in duodenal biopsy homogenates from non-T2D participants and patients with T2D.

Item	Non-T2D Patients (*n* = 15)	Patients with T2D (*n* = 20)	*p*-Value	Adjusted *p*-Value ^a^
Drug Metabolizing Enzymes				
CES-1	2.5 (1.3–3.7)	2.8 (2.1–3.8)	0.4	0.5
CES-2	2.1 (2.0–2.6)	2.3 (2.1–2.6)	0.8	0.9
CYP1A1	6.0 (1.5–15.2)	6.2 (3.3–20.2)	0.5	0.4
CYP2B6	4.1 (1.9–4.7)	4.8 (3.4–8.9)	0.3	0.2
CYP2C8	3.8 (1.8–6.2)	2.7 (1.4–4.4)	0.5	0.3
CYP2C9	9.0 (6.3–13.0)	11.9 (8.9–14.7)	0.051	0.09
CYP2C19	4.4 (3.3–5.2)	4.3 (3.4–5.6)	0.7	0.7
CYP2D6	2.4 (1.5–3.6)	2.9 (1.6–5.3)	0.1	0.1
CYP2E1	2.2 (0.9–6.8)	1.6 (1.2–2.6)	0.3	0.2
CYP2J2	1.8 (1.2–2.1)	2.0 (1.5–2.5)	0.3	0.3
CYP3A4	4.1 (2.9–5.9)	4.6 (3.5–7.1)	0.5	0.6
CYP3A5	3.9 (3.0–19.4)	4.2 (3.5–8.1)	0.3	0.3
Drug Transporters				
ABCB1	1.5 (1.1–2.0)	1.6 (1.2–1.9)	0.7	0.5
ABCG2	2.7 (2.3–3.6)	3.2 (2.8–4.1)	0.2	0.2
OATP2B1	0.7 (0.6–0.9)	0.9 (0.8–0.1)	0.02*	0.02 *

Results are expressed as median (interquartile range) of N-fold differences in drug metabolizing enzyme and transporter genes relative to the average expression of housekeeping genes and calibrator (2^−^^ΔΔ*C*T^). Each experiment was performed three times in triplicates. ^a^ Adjusted *p*-values from linear regression models analyses with diabetes status as a dichotomous predictor with sex and age as controls. * *p* < 0.05. Non-T2D, non-diabetic control patients group. T2D, patients with Type 2 diabetes group.

**Table 3 ijms-20-03257-t003:** Cytokine plasma levels in non-diabetic patients (Non-T2D) and patients with T2D.

Proinflammatory Cytokines	Non-T2D Patients (*n* = 16)	Patients with T2D (*n* = 20)	*p*-Value
IFN-γ	3.80 ± 2.67	3.41 ± 2.15	0.98
IL-1β	0.07 ± 0.07	0.09 ± 0.10	0.86
IL-6	0.99 ± 0.49	1.07 ± 0.94	0.35
TNF-α	2.00 ± 0.36	2.71 ± 1.25	0.03 *

Plasma concentrations (pg·mL^−1^) of proinflammatory cytokines are presented as mean ± SD. * *p* < 0.05. Non-T2D, non-diabetic control patient group. T2D, patients with type 2 diabetes group. IFN-γ, interferon-gamma. IL-1β, interleukine-1beta. IL-6, interleukine-6. TNF-α, tumor necrosis factor alpha.

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
