# Peer review of "A Pilot Study towards the Impact of Type 2 Diabetes on the Expression and Activities of Drug Metabolizing Enzymes and Transporters in Human Duodenum"

_ijms, 2019, doi:10.3390/ijms20133257_

Round 1
Reviewer 1 Report
The authors investigated the impact of type 2 diabetes on the expression of duodenum metabolizing enzymes, concluding that T2D do not modulate the expression of drug transporters/metabolizing enzymes in duodenum.
The work is potentially interesting, well planned, clear in explanation and with adequate laboratory methods. Its major limit is the small sample size, that in my opinion can affect statistical analysis, and as a result the conclusions
Author Response
We agree with the reviewer and to address the issue raised by the reviewer, a comment has been added in the discussion/conclusion section
Our data suggest that CYP450 activities were not modulated by T2D in the duodenum. This finding is based on a small number of subjects and an important intersubject variability was observed. Consequently, further investigations are needed to confirm our current findings and whether changes in reduced oral clearance observed clinically following oral drug administration in patients with T2D can be explained by a tissue-specific modulation occurring either in the liver or in other parts of the intestine.”
Reviewer 2 Report
In the manuscript “The impact of type 2 diabetes on the expression and activities of drug metabolizing enzymes and transporters in human duodenum” submitted by Gravel and colleagues, the authors measured the mRNA expression levels of drug metabolizing enzymes and transporters, the CYP450 activities, and the cytokines’ level in duodenal biopsies form patients with or without T2D. The conclusion is that T2D did not modulate the expression or activity of tested drug metabolizing enzymes and transporters in human duodenum. This study is well designed and well-written. A few little comments though for consideration:
1. Please revise the title of Table 1.
2. Are the subjects all Caucasian? The reviewer think Race is also a factor to investigate.
3. It is a pity that the subject number is small. Further trials are necessary in order to confirm this finding in the future.
4. The Discussion section: The reviewer would like to recommend a Table or Illustration to summarize the characters of CYP450s.
Author Response
Reviewer #2
1. Please revise the title of Table 1.
The title of Table 1 has been modified as follow: “Baseline demographic data and clinical characteristics of patients”
2. Are the subjects all Caucasians? The reviewer think Race is also a factor to investigate.
A comment has been added in the result section: “Most of the participants enrolled in our study were Caucasians (n=29), 2 subjects were Blacks and one Asian (data was missing for 4 individuals).”
We agree with the reviewer that ethnicity can influence the expression and activities of some CYP450 isoforms. Genotype analyses were performed in our study which is more accurate information than using race as a covariate. Even though some SNPs can be missing from our assays, the sample size is too small to detect any rare effect not included in our assay due the “race”. Most of the patients were Caucasians (29/36). In general, no difference in the activities were observed among our different ethnic groups (Caucasian Black, Asian and Other/not known) for midazolam, tolbutamide and ebastine (one-way ANOVA analysis, multiple comparison). The Table S4 shows the impact of genetic variants on CYP2C9, CYP2J2, CYP3A4/5 activities.
3. It is a pity that the subject number is small. Further trials are necessary in order to confirm this finding in the future.
We agree with the reviewers that further studies with a larger sample size are needed while looking at different parts of the intestine; this should be useful to better characterize the impact or not of diseases such as diabetes on CYP450 isoforms expressed in the intestine.
A comment has been added in the Discussion as follows: “Our data suggest that CYP450 activities were not modulated by T2D in the duodenum. This finding is based on a small number of subjects and an important intersubject variability was observed. Consequently, further investigations are needed to confirm our current findings and whether changes in reduced oral clearance observed clinically following oral drug administration in patients with T2D can be explained by a tissue-specific modulation occurring either in the liver or in other part of the intestine.”
4. The discussion section: The reviewer would like to recommend a Table or Illustration to summarize the characters of CYP450s.
The Appendixes 2-7 illustrate most of the data reviewed in the discussion section. Data have been presented in different Tables and Figures as supplemental materials since due to the amount and the diversity of data (mRNA expression, activities, correlation with inflammatory markers, genetic variants…).
Round 2
Reviewer 1 Report
I read the revised manuscript.
No further comments